# Characteristics and Health Risk Assessment of Semi-Volatile Organic Contaminants in Rural Pond Water of Hebei Province

**DOI:** 10.3390/ijerph16224481

**Published:** 2019-11-14

**Authors:** Lin Liu, Zhu Rao, Yuan Wang, Hamidreza Arandiyan, Jie Gong, Ming Liang, Feng Guo

**Affiliations:** 1National Research Center for Geoanalysis, Chinese Academy of Geological Sciences, Beijing 100037, China; liulinlucky@126.com (L.L.); fengguo@cags.ac.cn (F.G.); 2Key Laboratory of Ecological Geochemistry, Ministry of Natural Resources, Beijing 100037, China; 3School of Chemistry, Faculty of Science, The University of New South Wales, Sydney NSW 2052, Australia; Yuan.wang4@unsw.edu.au; 4Laboratory of Advanced Catalysis for Sustainability, School of Chemistry, The University of Sydney, Sydney 2006, Australia; hamid.arandiyan@sydney.edu.au; 5College of Geoscience and Surveying Engineering, China University of Mining and Technology, Beijing 100083, China; gongj117286@163.com (J.G.); mingliang0918@163.com (M.L.)

**Keywords:** semi-volatile organic contaminants, health risk assessment for human, rural pond water

## Abstract

Pond water as surface water has certain environmental impacts on environmental media such as groundwater, lakes, atmosphere, and soil. Organic pollutants present in pond water may pose health risks to humans, but research on organic pollutants in pond water is rare. Here, taking pond water collected in rural areas of Hebei province as the sample, we analyzed and evaluated four categories of semi-volatile organic compounds (SVOCs), including 11 phenolic compounds, 7 aniline compounds, 16 parent polycyclic aromatic hydrocarbons (PAHs), 14 PAH derivatives, and 16 phthalate esters (PAEs). The results show that the 10 water samples contained 26.2–17034 ng/L of Σ phenols, 33.7–2612 ng/L of Σ anilines, 33.9–1651 ng/L of Σ PAHs, and 59.0–2800 ng/L of Σ PAEs. Furthermore, non-carcinogenic risk and carcinogenic risk caused by SVOCs through direct ingestion and dermal exposure were also assessed. The current levels of non-carcinogenic risks and carcinogenic risks through these two means of exposure are within acceptable limits, except for the site 1 and site 5 in Hebei province where a total cancer risk exceeds 10^−6^. It can be concluded that the pond water studied had a low risk of carcinogenicity to the human.

## 1. Introduction

Investigation of water contaminants such as organic contaminants and heavy metals has become the focus of environmental research. High concentrations of heavy metals are considered highly toxic for humans and aquatic life. The sources of heavy metals in surface water may originate from geogenic or anthropogenic activities, such as weathering of bedrocks, sulfide deposits, mining, industry, and agricultural activities [1]. Organic contaminants and heavy metals in water can both cause a potential risk to human health. Some studies have indicated the risk of heavy metals in water to human health, as well as the source of those heavy metals and their formation in minerals [1,2,3]. In this study, the aim is to analyze the organic contaminants in water. Semi-volatile organic contaminants (SVOCs) are persistent pollutants in aquatic environments, soils, and sediments that are highly toxic and difficult to decompose. They can be transported over long distances and can be accumulated in organisms [4,5]. Most SVOCs can cause cancer, reproductive disorders, nervous system damage, and immune system disruption [6]. Polycyclic aromatic hydrocarbons (PAHs), a major group of SVOCs, pose a significant threat to environmental and human health due to their persistence, carcinogenicity, mutagenicity, and toxicity [7,8]. The United States Environmental Protection Agency (USEPA) lists 16 PAHs as priority pollutants, 7 of which pose a cancer risk to human health. PAH derivatives are more toxic than PAH precursors. Some PAHs can also cause endocrine disruption and affect the immune system [8]. The sources of PAHs are primarily considered to be anthropogenic activities such as fossil fuel production and combustion [7,8].

Phthalic acid esters (PAEs) are ubiquitous environmental pollutants. PAEs have been used in a variety of daily life products such as pesticide carriers, insect repellents, cosmetics, fragrances, and lubricant, but they are mainly used as plasticizers for plastics [5] and can be migrated from plastic to external environments under certain conditions [9]. PAEs, such as diisobutyl phthalate (DIBP), can disrupt the endocrine systems of animals and destroy the reproductive systems of female animals [9,10]. Furthermore, some studies have shown that di-(2-ethylhexyl) phthalate (DEHP) exposure can increase Leydig cell numbers in rats, but inhibit the differentiation process during regeneration [11]. The USEPA has identified these as potentially harmful to human health.

Aniline derivatives (ADs) are toxic, and special colors and odors may cause carcinogenic effects on human health and have been prescribed as priority pollutants in China. ADs are widely used as raw materials or intermediates in industry synthesis [12,13,14]. Long-term exposure to aniline compounds may cause a loss of hemoglobin and produce anemia [15]. Aniline may enter the environment as an intermediate in dye synthesis or in a commercial application such as pharmaceuticals, photographic developers, varnish, and perfumes [13,16]. The exposure pathway of aniline is mainly through inhalation of steam or by absorption of liquid, and aniline is easily absorbed into the body through the skin [13].

Phenolic compounds are common organic pollutants in water and are easily absorbed by the skin [17]. The key sources of phenolic compounds in the environment are agricultural and industrial activities such as coke ovens, oil refineries, dyes, pharmaceuticals, mining, and insecticides [18,19,20,21,22]. The impacts of phenolic compounds on human health are damage to organs, tissues, and the nervous system [17,18,19].

SVOCs in water can cause a potential risk to human health and the environment, so it is important to assess the risk of organic pollutants in water. Some studies have shown that the non-carcinogenic risk of organic pollutants in water can be neglected, but certain cancer risks for individual pollutants do exist [4,5,6,23]. For example, individual organic compounds of drinking water in Huaihe River have posed a relatively high cancer risk to human health [24]. The method for human health risk assessment recommended by the USEPA has been applied to determine the potential adverse effects posed on humans by pollutants [24,25], but the hazards posed by chemical mixtures are challenging to evaluate because the toxicity pathways of many chemicals are not known. Considering the worst cases, some studies have calculated the sum of hazard quotients for each compound, to estimate the combined risks of mixture components [26,27].

Previous studies have mainly focused on the distribution and health risk assessment of SVOCs in the main surface water basins, groundwater, and drinking water of China [4,5,23,26,28,29]. However, the pollutant profiles of rural pond water have been ignored. Most ponds are formed by artificial excavation and can be used in industry and domestic and agricultural activities. However, as the economy develops, these ponds become a place to dump garbage and discharge sewage. In the long run, due to weak self-purification capacities, the ponds will become dead ponds with no live water supply, causing environmental degradation and water pollution. In general, ponds are not only temporary “sinks” from various pollutants such as villages and farmland, but also “sources” of large-scale water pollutants in basins. Moreover, as time goes by, organic pollutants in ponds gradually accumulate, and the concentration increases, increasing the risk of human exposure and posing a threat to human and animal health [4]. High levels of organic pollutants in ponds can cause various diseases such as cancer, femoral head necrosis, damage to the nervous system, and disruption of the immune system through washing, bathing, and consuming sewage. Organic contaminants in pond water exhibit high persistence and definite mutagenic potential [30]. In addition, pond water may also affect the innate immune responses of shrimp and increase the incidence of amphibian hatchling malformations when interacting with UV radiation [31,32]. Pond water in study area is ubiquitous, and the study area is close to the Baiyangdian Lake—an important water source for residents. Therefore, it is necessary to investigate the organic pollutants in ponds of the study area. In this study, the levels of SVOCs contamination in rural pond water were determined. Moreover, a preliminary study on the effects of detected contaminants on human health through water consumption were conducted according to the standards of the USEPA [4].

## 2. Materials and Methods

### 2.1. Sample Collection

A total of 10 pond water samples were collected from rural areas of Hebei province in Eastern China during 2017. Figure 1 shows the 10 sampling sites, which were located in residential areas without factories, domestic sewage discharge, or sewage treatment plants. Hence, there may be a certain amount of semi-volatile organic contaminants in these areas. During the sampling process, some water parameters, such as pH, water temperature salinity, and solid solubility, were detected at the sites. Water samples were collected in 1-L amber glass bottles and preserved in ice-filled boxes until being transferred to the laboratory and stored in a 4 °C refrigerator for further analysis.

### 2.2. Chemical Analysis and Sample Pretreatment

A total of 64 semi-volatile organic contaminants from 10 water samples were measured and classified into 4 categories, including 11 phenolic compounds, 7 anilines compounds, 16 phthalates, 16 parent polycyclic aromatic hydrocarbons, and 14 PAH derivatives. In this study, the reagents used were all pesticide residue analysis grades. Six internal standards (1,4-dichlorobenzene-d4, naphthalene-d8, acenaphthene-d10, chrysene-d12, phenanthrene-d10 and perylene-d12) with concentrations of 4000 μg/mL and surrogate standards (p-terphenyl-d14 and 2-fluorobiphenyl) with concentrations of 200 μg/mL were purchased from Accustandard (New Haven, CT, USA).

Sample extraction was carried out using the liquid-liquid extraction method described elsewhere [33]. The process is described as follow: 1 L of water was added into a 1 L separatory funnel while adding 30.0 g of NaCl and 30 μL of surrogate standards (10.0 μg/mL). Then, 50 mL of dichloromethane (J&K Scientific, Beijing China) was added to the funnel. The funnel was shaken vigorously for 10 min. After phase separation, the organic phase was separated into a 250-mL flat-bottomed flask. The same extraction process was repeated twice, and all target extractants were collected in 30 mL of dichloromethane at various pH values (pH > 11 and pH < 2). An appropriate amount of anhydrous sodium sulfate was added to remove water. Each combined extract was rotary evaporated to concentrate the sample into a volume of 2-3 mL. The extract samples were concentrated into approximately 1 mL under a gentle nitrogen stream, with 30 μL of internal standards (10.0 μg/mL) added to each sample before adjusting the volume of the extract to 1 mL.

All water samples were analyzed using the gas chromatography coupled with mass spectrometry (GCMS-TQ8040, SHIMADZU, Kyoto Japan) and a DB-5MS silica-fused capillary column (30 m × 250 μm × 0.25 μm). The carrier gas was helium (99.999%), and the flow rate was 1.5 mL/min. For the target compounds, the column temperature program was: an initial temperature of 60 °C, held for 3 min, then ramped to 270 °C at 10 °C/min for 1 min, raised to 285 °C at 5 °C/min, and finally ramped to 310 °C at 10 °C/min for 5 min. The injection temperature was 270 °C and the injection volume of samples was 1 μL in the splitless mode. The acquisition method was a selected ion monitoring (SIM) mode.

### 2.3. Quality Assurance and Quality Control

The quantification method for the target compounds was as follows: The peak areas of the target compounds were normalized using the internal standards and quantified using the five-point calibration curve with an R^2^ value greater than 0.99. To ensure the accuracy of the experimental process and the recovery of the target compounds, a procedure blank, spiked blanks, and duplicate samples were conducted on about 10% of all samples according to standard operating procedures. Procedure blanks were adopted to exclude contamination and interferences during the whole procedure. All the samples were spiked with known amounts of surrogate standards prior to pretreatment. The recoveries of the target compounds were 50–130%, and the recoveries of all the surrogate standards were 70–110%. The detection limits of contaminants ranged from 2.0 to 20 ng/L.

### 2.4. Human Risk Assessment

Human health risk assessments were carried out based on reliable exposure pathways for contaminants [34]. Contaminants in the atmosphere, water, soil, and food chain enter the human body through direct ingestion, incidental inhalation, and dermal absorption [6,35]. Occasional intake is in trace amounts, so this study mainly evaluated the harm caused by organic pollutants entering the human body in pond water through direct ingestion and dermal absorption. The exposure doses were calculated using Equations (1) and (2), respectively, which were obtained from the USEPA [34,36]:(1)$$Di=\frac{C_{w}\times IR\times EF\times ED}{BW\times AT}$$
(2)$$Dd=\frac{C_{w}\times Kp\times SA\times ET\times EF\times ED}{BW\times AT}$$where *Di* (mg/kg·day) is the intake dose from the ingested water; *Dd* (mg/kg·day) is intake dose from dermal absorption; *Cw* (mg/L) is the average concentration of organic pollutants in water; *IR* (L/day) is the ingestion rate (2 L/d); *EF* (day/year) is the exposure frequency (365 days/years); *ED* (year) is the exposure duration (70 years [37]); *BW* (kg) is the average body weight (for Chinese adults, the BW value is 60 kg [4]); *AT* (day) is the averaged time (70 years × 365 days/years); *SA* (cm^2^) is the exposed skin surface area (16,000 cm^2^ [26]); *ET* (hours/day) is the exposure time (0.2 h/day [37]); and *Kp* (cm/h) is the dermal permeability constant. It is necessary to understand the fact that the real exposure frequency of humans to organic pollutants following the above two exposure pathways will not be 365 days.

According to the carcinogenicity of organic pollutants, health risk assessment includes both carcinogenic and noncarcinogenic health hazard indices. The hazard quotient (*HQ*) was calculated using Equation (3) to evaluate noncarcinogenic risks [34]:(3)HQ=E/RfDwhere *E* (mg/kg·day) represents the exposure dose obtained from Equations (1) and (2), which is equivalent to *Di*; *RfD* (mg/kg·day) is the oral reference dose of the toxicant.

Carcinogenic risk is divided into low-dose cancer risk levels and high carcinogenic risk levels. The equation is as follows:(4)low-dose carcinogenic risk:Risk=CDI×SF
(5)high carcinogenic risk: Risk=1−exp(−CDI×SF)when the low-dose risk value is higher than 0.01, Equation (5) is used to calculate the risk of carcinogenesis. *CDI* (mg/kg· day) is the chronic daily intake averaged over 70 years as identified from Equations (1) and (2), and *SF* (mg/kg·day)^−1^ is the slope factor. The values of *RfD*, *SF*, and *Kp* are gained from the Integrated Risk Information System (IRIS) [23] and listed in Table 1.

The pollution in the environment is mostly compound pollution. It is generally believed that the risks caused by each pollutant are additive, not synergistic or antagonistic. So, the noncarcinogenic hazard index and carcinogenic risk index of all pollutants is equal to the sum of the hazard quotient of each pollutant:(6)HI=HQ1+HQ2+⋯+HQn
(7)∑R=R1+R2+⋯+Rnwhere *HQn* represents the hazard quotient for the n-th toxicant, *Rn* represents the risk index of the n-th toxicant. When the noncarcinogenic hazard index (*HI*) is less than 1, it is considered that there is no risk; when the *HI* is greater than 1, it indicates that there is a considerable hazard. For carcinogenic risk, a risk value lower than 10^−6^ is considered acceptable, a risk value between 10^−6^ and 10^−4^ represents a certain degree of risk but is acceptable [26], and a risk value exceeding 10^−4^ indicates a significant risk for local residents [4].

## 3. Results and Discussion

### 3.1. Characteristics of Semi-Volatile Organic Contaminants in Rural Pond Water

Twenty-seven pollutants of the 64 target pollutants were detected in 10 pond water samples with a total concentration of 207–18,713 ng/L. The contaminants detected included 6 phenols, 2 anilines, 10 parent PAHs, 6 PAHs derivatives and 4 PAEs (Table 2). Six compounds were detected in the 10 pond water samples at a detection rate of not less than 80%, including benzo[a]anthracene (100%), naphthalene (90%), fluoranthene (90%), pyrene (80%), chrysene (80%), and benzo[b]fluoranthene (80%). The total concentration of organic pollutants detected in pond water at the S1, S5, and S8 sample points exceeded 1000 ng/L, and the highest concentration appeared for S5 (18,713 ng/L), where 13 compounds were detected (Appendix A). For S5, phenol, m-cresol, and p-cresol were the dominant pollutants, accounting for 91% of the total concentration. Compared with other surveyed rivers in China, the concentrations of detectable SVOCs in this study (3113 ng/L) were lower than those found in the Yangtze (5344.5 ng/L) [28] and Wujin (24,845 ng/L) rivers [25], but they were higher than the Lhasa River (1980 ng/L) [6] and Yellow Sea (204.24 ng/L) [23]. This indicates that the pond water in this studied area might have contained a medium level of SVOCs. Figure 2 shows the distribution and total concentration of all detected organic pollutants at each sampling site.

By analyzing the target of 11 phenols, at least 6 phenolic compounds were detected, namely phenol, m-cresol, p-cresol, o-nitrophenol, 2,4-dimethylphenol, and 2,4-dichlorophenol (Table 2). Among them, phenol, m-cresol, and 2,4-dichlorophenol have been listed as priority pollutants in China [22]. As can be seen in Figure 2, phenolic compounds were detected in five water samples, including points S1, S2, S5, S8, and S9, with a total concentration of 26.2–17,034 ng/L. The highest total phenol concentration appeared at point S5, and four phenols compounds were detected, including phenol (2614 ng/L), m-cresol (1544 ng/L), p-cresol (12,860 ng/L), and 2,4-dichlorophenol (16.1 ng/L), which was more than 10 times that of other sampling sites (Figure 3). Four phenol components were detected at point S8, including phenol (71.9 ng/L), m-cresol (418 ng/L), 2,4-dimethyphenol (155 ng/L), and 2,4-dichlorophenol (191 ng/L). Phenol of the detected phenolic compounds has the highest detectable rate in 10 pond water samples (50%), with concentrations ranging from 26.2 to 2614 ng/L (Table 2). The highest concentration of phenol (2614 ng/L) exceeded the 2000-ng/L drinking water limit in China. The concentration of p-cresol at S6 reached 12,860 ng/L. Among the phenolic compounds detected, 2,4-dichlorophenol is more toxic than other phenols. The limit of 2,4-dichlorophenol in the “Environmental quality standards for surface water of China (GB3838-2002)” is 93 μg/L. However, the maximum concentration of 2,4-dichlorophenol in the 10 samples was 36.0 ng/L, which was much lower than the standard value. Compared with other studies, the concentration of total phenolic compounds in rural pond water was lower than that of the tributaries of the Hun River [38], but higher than Taihu Lake [39] and the rainy season surface water of Three Gorges Reservoir [40]. These results indicate the existence of point source pollution characteristics, and the phenols in rural pond water may have originated from domestic sewage, domestic garbage, and industrial wastewater. It is necessary to strengthen the supervision and proper disposal of water bodies.

For seven aniline compounds, only aniline and p-chloroaniline were detected (Table 2). The total concentration ranged from 33.7 to 2612 ng/L. As shown in Figure 2, aniline was detected in 50% of pond water samples at a concentration ranging from 33.7 to 2572 ng/L. The highest concentration of aniline found at S8 was 2572 ng/L, which was much lower than the standard value of the “Environmental quality standards for surface water of China (GB3838-2002)” (0.1 mg/L). P-chloroaniline was observed at three sampling sites (S6, S7, and S8), at concentrations between 39.9 and 65.0 ng/L. At the same time, a sinkhole was found near the S8, indicating that the water near S8 was contaminated to a certain extent, and that further risk assessment is required.

For the 30 targets PAHs, 15 PAHs were detected at least once, including 10 parent PAHs and 5 PAH derivatives with total concentrations of 33.9-1651 ng/L (Figure 2). As seen in Figure 4, 15 PAHs were detected in all samples, and at least one PAH derivative was found in five samples (S1, S2, S7, S8, and S10, respectively). The highest total concentration was at S1, reaching 1651 ng/L, where 8 compounds were detected. At S1, 1-aminonaphthalene and 2-aminonaphthalene were the dominant contaminants, accounting for 95% of the total concentration (Figure 4). Compounds 2-methylnaphthalene, 1-methylnaphthalene, and 1,3-dimethylnaphthalene were detected only at S8, with concentrations of 82.1, 92.0, and 40.4 ng/L, respectively.

At least one parent PAH was detected at all sampling sites, with total concentrations ranging from 33.9 to 369 ng/L (Figure 4). Ten parent PAHs were priority PAHs of the USEPA, including naphthalene (NAP), acenaphthylene (ACY), fluorene (FLO), phenanthrene (PHE), fluoranthene (FLA), pyrene (PYR), benzo[a]anthracene (BaA), chrysene (CHR), benzo[b]fluoranthene (BbF), and benzo[k]fluoranthene (BkF). BaA was the most frequently detected compound, with concentrations ranging from 5.04 to 42.5 ng/L (Table 2). The maximum concentration of BaA (42.5 ng/L) was lower than the standard value (275 ng/L) [25]. The highest total concentration of parent PAHs was at S5, where seven compounds were detected. The second-highest total concentration was the detection of eight compounds at S6 (245 ng/L), followed by the detection of eight compounds at S2 (202 ng/L). In addition, eight compounds were also detected at both S7 (144 ng/L) and S9 (62.8 ng/L). To further understand the status of PAHs in rural pond water, the levels of PAHs in this study were compared with other surveyed rivers. The mean concentration of Σ PAHs in this study (374 ng/L) was slightly higher than that of the Lhasa (330 ng/L) [6] and Yellow (248 ng/L) rivers [41], but lower than most other rivers, including the Yangtze (2095 ng/L) [42], Wujin (2133 ng/L) [25], Hun (4347 ng/L) [43], and Tianjin (14,000 ng/L) [44]. The results indicate that the PAH levels in pond water were not high.

The main sources of parent PAHs are combustion and petroleum [45]. The source of PAHs can be inferred from the ratio of the parent PAHs isomers, such as Phe/(Phe + Ant), Flu/(Flu + Pyr), and BaA/(BaA + Chr). In this study, the isomer ratio of Flu/(Flu + Pyr) was used to speculate the source of PAHs [8,46,47]. The Flu/(Flu + Pyr) of all samples was below 0.5, indicating that gasoline, diesel, bituminous coal combustion, and emissions from cars and diesel trucks were the predominant sources of PAHs [46].

It can be seen from the above results that there was not a large amount of PAH pollution input in the 10 pond water samples, and the pollution was not severe, but pollutants were generally detected. No carcinogenic benzo[a]pyrene was detected at 10 sample points, so the risk of PAHs was small. The distribution of 10 pond water samples was relatively distant and relatively independent.

PAEs are widely present in the surroundings. Unsurprisingly, PAEs were detected at 60% of the sampling sites (Figure 2). Four PAEs were found with the total concentrations between 59 and 2800 ng/L (Figure 2). Among them, DMP and DEHP are the priority pollutants of the USEPA. As shown in Figure 2, the highest total concentration of PAEs appeared at S8 (2800 ng/L), with dimethyl phthalate (DMP) being the dominant compound. Subsequently, the second-highest concentration was at S5, with a total concentration of 1310 ng/L. Diisobutyl phthalate (DIBP) and dibutoxyethyl phthalate (DBEP) were found only at S8 and S5, at concentrations of 630 and 350 ng/L, respectively. In four samples, di-(2-ethylhexyl) phthalate (DEHP) was detected with a median concentration of 708 ng/L (Table 2). Compared with other rivers in China, the PAE levels in rural pond water were much lower than those in the Wuhan section of the Yangtze River during low water periods (36-91 μg/L) [48], as well as the Beijiang (4900–20,000 ng/L) [5] and Wujin (22,046 ng/L) rivers [25], but close to those in the Yellow River estuary (1011 ng/L) [49]. The comparisons suggest that the PAE levels in rural pond water were relatively low.

### 3.2. Human Health Risk Assessment

The health risk assessment of water consumption is an effective method for investigating the hazards caused by contaminants [4]. The carcinogenic and noncarcinogenic risks of four groups of SVOCs in each pond water sample were evaluated (Appendix A). Since *RfD* and *SF* are not available for all detected semi-volatile organic pollutants [6], only the noncarcinogenic risks of 14 organic pollutants and the carcinogenic risks of 8 organic pollutants were evaluated. Table 3 shows the noncarcinogenic risk and the carcinogenic risk quotients for adults exposed to organic pollutants in pond water samples through different pathways (direct ingestion and dermal absorption). The highest *HI* value for noncarcinogenic risk resulting from ingestion and dermal exposure was lower than 1. However, the highest *HI* for carcinogenic risk in both pathways was higher than 10^−6^. The total risk of each sampling site is shown in Figure 5. For noncarcinogenic risk, as shown in Figure 5a, the total risk via two exposure pathways ranged from 5.18 × 10^−5^ to 1.42 × 10^−2^, and the *HI* ranges through the direct ingestion pathway and dermal absorption pathway in 10 pond water samples were 4.51 × 10^−5^–1.41 × 10^−2^ and 6.67 × 10^−6^–3.01 × 10^−3^, respectively (Appendix A). These results indicate that the *HI* values of these contaminants in all samples were less than 1, and their non-cancer impacts on human health can be neglected.

Considering the risk of carcinogenesis, the total cancer risks from two exposure pathways ranges were 9.51 × 10^−8^–1.81 × 10^−6^, and the *∑R* ranges from the direct ingestion pathway and dermal absorption pathway were 5.34 × 10^−8^–8.99 × 10^−7^ and 4.17 × 10^−8^–1.05 × 10^−6^, respectively (Appendix A). As shown in Figure 5b, the total carcinogenic risks for all sites were lower than 10^−6^, expect for S1 and S5. The carcinogenic risk at S1 and S5 were higher than 10^−6^, but lower than 10^−4^. DEHP at S1 and S5 was identified as the prominent contributor of carcinogenic risk. For a single component, the noncarcinogenic *HQ* caused by direct ingestion was at least one order of magnitude higher than that of the dermal absorption pathway, expect for 2-methylnaphthalene and DEHP (Table 3). However, for the carcinogenic risk value of single compounds, the *HQ* of the corresponding component in the two exposure pathways was still of the same order of magnitude, except for the *HQ* of aniline and p–chloroaniline caused by the direct ingestion pathway, which were much higher than that caused by dermal absorption. In general, for the 10 pond samples, organic pollutants entering the human body through direct ingestion and dermal absorption do not pose significant harm to human health.

The human health risk levels of organic contaminants in rural pond water are similar to other studies [5,6,23,26]. Therefore, this assessment method can reflect, to some extent, the harm of water bodies in the region to human health, which may be a very useful tool in revealing the true meaning and relevance of organic pollutants [6].

## 4. Conclusions

This study analyzed and evaluated 64 semi-volatile organic contaminants in 10 rural pond water samples collected in Hebei province, and evaluated the *HQ* and *HI* values through the direct ingestion and dermal absorption pathways to assess the potential hazards to humans. The concentrations of phenol, aniline, and PAHs in water were very low compared to PAEs, and in most samples the concentrations of organic contaminants were also low, except at individual points such as S1 and S8. The levels of SVOCs in rural pond water were moderate compared to other studies. As for the health risk assessment results, the noncarcinogenic and carcinogenic risks via direct ingestion and dermal absorption were within acceptable limits at most sampling sites. At S5, organic contaminants in pond water via showering and bathing may pose certain carcinogenic risks to the residents. Therefore, it is necessary to pay attention to the health hazards of organic pollutants in pond water. This investigation of organic pollution in rural pond water can provide a basis for surface water pollution control in the future, and hopefully will attract people’s attention to pond water pollution. It is noteworthy that there were some limitations and uncertainties in certain methods of health risk assessment of SVOCs in this study, such as the assumption of exposure frequency and possible interaction among chemicals, which could vary in different situations. There is much work to be done in improving methods of the human health risk assessment.

## Figures and Tables

**Figure 1 ijerph-16-04481-f001:**
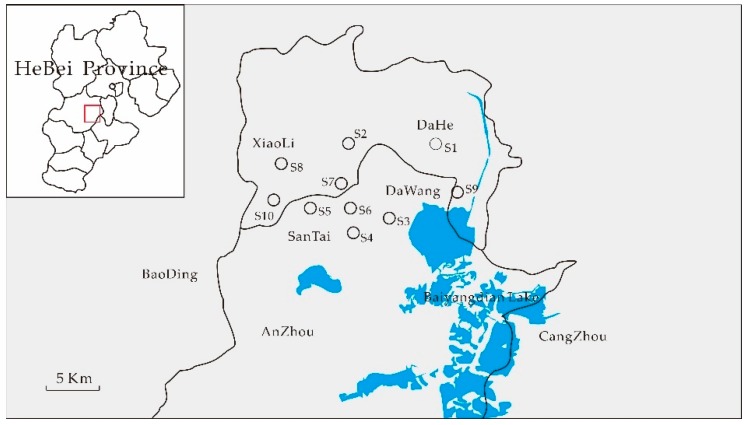
Sampling sites in rural areas of Hebei province.

**Figure 2 ijerph-16-04481-f002:**
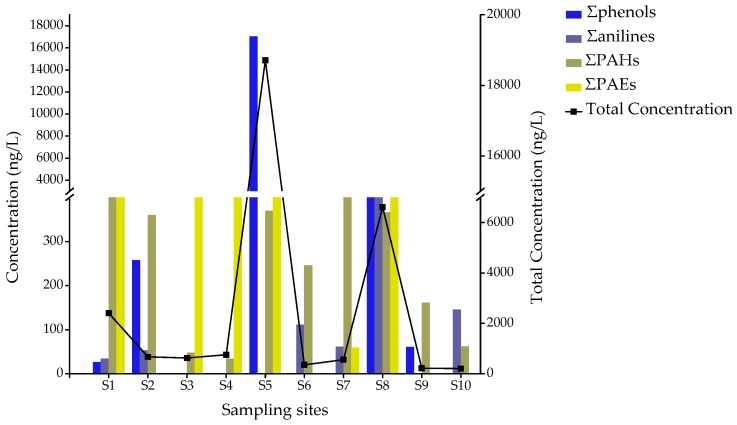
Distribution and total concentration of all detected compounds in each pond water sample of Hebei province.

**Figure 3 ijerph-16-04481-f003:**
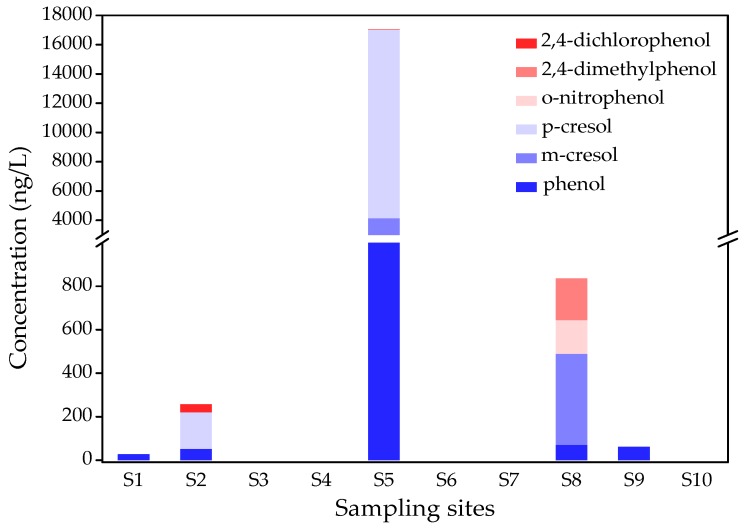
Concentration and distribution of phenol compounds in the 10 pond water samples of Hebei province.

**Figure 4 ijerph-16-04481-f004:**
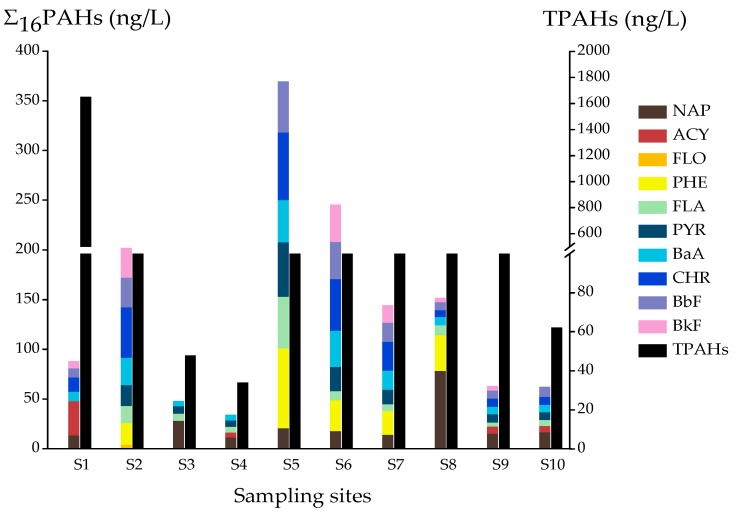
Concentrations and distribution of Total PAHs and 16 PAHs in 10 pond water samples of Hebei province.

**Figure 5 ijerph-16-04481-f005:**
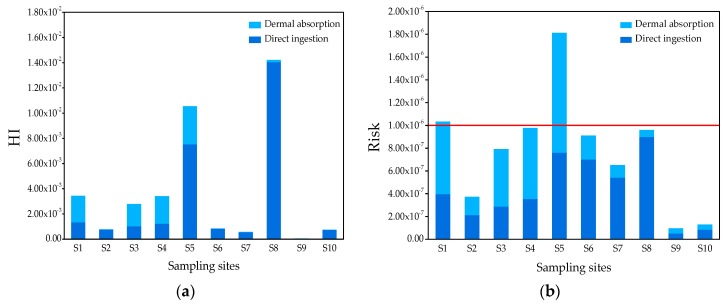
(**a**) Noncarcinogenic risk of organic pollutants detected in the pond water samples. (**b**) Carcinogenic risk of organic pollutants in 10 pond water samples. The red line represents the limitation of carcinogenic risk.

**Table 1 ijerph-16-04481-t001:** The values of the oral reference dose (*RfD*), dermal permeability constant (*Kp*), and slope factor (*SF*) parameters for carcinogenic and noncarcinogenic assessment.

Compounds	*RfD* (mg/kg Day)	Kp (cm/h)	*SF* (mg/kg·Day)^−1^
phenol	3.00 × 10^−1^	4.3 × 10^−3^	-
m-cresol	5.00 × 10^−2^	7.8 × 10^−3^	-
p-cresol	1.00 × 10^−1^	7.5 × 10^−3^	-
o-nitrophenol	-	-	-
2,4-dimethylphenol	2.00 × 10^−2^	1.1 × 10^−2^	-
2,4-dichlorophenol	3.00 × 10^−3^	2.1 × 10^−2^	-
aniline	7.00 × 10^−3^	1.9 × 10^−3^	5.70 × 10^−3^
p-chloroaniline	4.00 × 10^−3^	5.0 × 10^−3^	2.00 × 10^−1^
naphthalene	2.00 × 10^−2^	4.7 × 10^−2^	-
2-methylnaphthalene	4.00 × 10^−3^	9.2 × 10^−2^	-
1-methylnaphthalene	7.00 × 10^−2^	9.3 × 10^−2^	2.90 × 10^−2^
1,3-dimethylnaphthalene	-	-	-
acenaphthylene	-	-	-
1-aminonaphthalene	-	-	-
2-aminonaphthalene	-	-	-
fluorene	4.00 × 10^−2^	1.1 × 10^−1^	-
phananthrene	-	1.4 × 10^−1^	-
fluoranthene	4.00 × 10^−2^	3.1 × 10^−1^	-
pyrene	3.00 × 10^−2^	2.0 × 10^−1^	-
benzo[a]anthracene	-	5.5 × 10^−1^	1.00 × 10^−1^
chrysene	-	6.0 × 10^−1^	1.00 × 10^−3^
benzo[b]fluoranthene	-	4.2 × 10^−1^	1.00 × 10^−1^
benzo[k]fluoranthene	-	6.9 × 10^−1^	1.00 × 10^−2^
dimethyl phthalate	-	1.4 × 10^−3^	-
diisobutyl phthalate	-	-	-
dibutoxyethyl phthalate	-	-	-
di-(2-ethylhexyl) phthalate	2.00 × 10^−2^	1.1	1.40 × 10^−2^

Note: “-” stands for not available.

**Table 2 ijerph-16-04481-t002:** Minimum (Min), maximum (Max), mean, and median concentrations, with the standard deviations (SD) and detected number (DN) of detected contaminants in 10 rural pond water samples (ng/L).

Compounds	Min	Max	Mean	Median	SD	DN
**Phenolic Compounds**						
phenol	26.2	2614	565	60.8	1145	5
m-cresol	418	1544	981	981	796	2
p-cresol	168	12860	6514	6514	8975	2
o-nitrophenol	155	155	155	155	-	1
2,4-dimethylphenol	191	191	191	191	-	1
2,4-dichlorophenol	16.1	36.0	26.1	26.1	14.0	2
**Anilines**						
aniline	33.7	2572	570	52.7	1120	5
p-chloroaniline	39.9	65.0	55.4	61.1	13.5	3
**PAHs ^a^ and their Derivatives**						
naphthalene	11.5	78.5	24.0	16.7	21.0	9
2-methylnaphthalene	82.1	82.1	82.1	82.1	-	1
1-methylnaphthalene	92.0	92.0	92.0	92.0	-	1
1,3-dimethylnaphthalene	40.4	40.4	40.4	40.4	-	1
acenaphthylene	5.07	34.5	13.2	6.65	14.2	4
1-aminonaphthalene	28.7	826	236	45.3	393	4
2-aminonaphthalene	69.6	737	292	181	305	4
fluorene	3.98	3.98	3.98	3.98	-	1
phenanthrene	22.0	80.6	38.6	31.1	24.1	5
fluoranthene	4.29	52.1	13.2	7.07	15.1	9
pyrene	6.48	54.6	18.0	11.4	16.2	8
benzo[a]anthracene	5.04	42.5	16.9	8.9	13.9	10
chrysene	7.11	68.2	29.7	21.8	24.2	8
benzo[b]fluoranthene	7.82	51.0	21.5	14.5	16.3	8
benzo[k]fluoranthene	4.24	37.3	16.6	12.1	14.1	6
**PAEs ^b^**						
dimethyl phthalate	59.0	2170	1115	1115	1493	2
diisobutyl phthalate	630	630	630	630	-	1
dibutoxyethyl phthalate	350	350	350	350	-	1
di-(2-ethylhexyl) phthalate	580	960	739	708	160	4

Note: “-” represents below the detection limit. Undetected contaminants were not included; “a” represents polycyclic aromatic hydrocarbons; “b” represents phthalate esters.

**Table 3 ijerph-16-04481-t003:** The min, mean, and max of hazard quotient (*HQ*) and hazard index (*HI*) for non-carcinogenic risk and carcinogenic risk from direct ingestion and dermal absorption.

**Non-Carcinogenic Risk**						
**Compounds**	**Direct Ingestion**			**Dermal Absorption**		
	**Min**	**Mean**	**Max**	**Min**	**Mean**	**Max**
phenol	2.91 × 10^−6^	6.28 × 10^−5^	2.90 × 10^−4^	2.00 × 10^−8^	4.32 × 10^−7^	2.00 × 10^−6^
m-cresol	2.79 × 10^−4^	6.54 × 10^−4^	1.03 × 10^−3^	3.47 × 10^−6^	8.13 × 10^−6^	1.28 × 10^−5^
p-cresol	5.60 × 10^−5^	2.17 × 10^−3^	4.29 × 10^−3^	6.76 × 10^−7^	2.62 × 10^−5^	5.17 × 10^−5^
2,4-dimethylphenol	3.18 × 10^−4^	3.18 × 10^−4^	3.18 × 10^−4^	5.54 × 10^−6^	5.54 × 10^−6^	5.54 × 10^−6^
2,4-dichlorophenol	1.79 × 10^−4^	2.90 × 10^−4^	4.00 × 10^−4^	5.91 × 10^−6^	9.55 × 10^−6^	1.32 × 10^−5^
aniline	1.61 × 10^−4^	2.71 × 10^−3^	1.22 × 10^−2^	4.88 × 10^−7^	8.25 × 10^−6^	3.72 × 10^−5^
p-chloroaniline	3.33 × 10^−4^	4.61 × 10^−4^	5.42 × 10^−4^	2.64 × 10^−6^	3.66 × 10^−6^	4.30 × 10^−6^
naphthalene	1.91 × 10^−5^	4.00 × 10^−5^	1.31 × 10^−4^	1.43 × 10^−6^	2.98 × 10^−6^	9.76 × 10^−6^
2-methylnaphthalene	6.84 × 10^−4^	6.84 × 10^−4^	6.84 × 10^−4^	1.00 × 10^−4^	1.00 × 10^−4^	1.00 × 10^−4^
1-methylnaphthalene	4.38 × 10^−5^	4.38 × 10^−5^	4.38 × 10^−5^	6.53 × 10^−6^	6.53 × 10^−6^	6.53 × 10^−6^
fluorene	3.32 × 10^−6^	3.32 × 10^−6^	3.32 × 10^−6^	5.84 × 10^−7^	5.84 × 10^−7^	5.84 × 10^−7^
fluoranthene	3.58 × 10^−6^	1.10 × 10^−5^	4.34 × 10^−5^	1.76 × 10^−6^	5.41 × 10^−6^	2.14 × 10^−5^
pyrene	7.20 × 10^−6^	2.00 × 10^−5^	6.06 × 10^−5^	2.32 × 10^−6^	6.44 × 10^−6^	1.95 × 10^−5^
di-(2-ethylhexyl) phthalate	9.67 × 10^−4^	1.23 × 10^−3^	1.60 × 10^−3^	1.75 × 10^−3^	2.23 × 10^−3^	2.89 × 10^−3^
*HI*	3.06 × 10^−3^	8.71 × 10^−3^	2.17 × 10^−2^	1.88 × 10^−3^	2.41 × 10^−3^	3.18 × 10^−3^
**Carcinogenic Risk**						
**Compounds**	**Direct Ingestion**			**Dermal Absorption**		
	**Min**	**Mean**	**Max**	**Min**	**Mean**	**Max**
aniline	6.41 × 10^−9^	1.08 × 10^−7^	4.89 × 10^−7^	1.95 × 10^−11^	3.29 × 10^−10^	1.49 × 10^−9^
p-chloroaniline	2.66 × 10^−7^	3.69 × 10-7	4.34 × 10^−7^	2.11 × 10^−9^	2.93 × 10^−9^	3.44 × 10^−9^
1-methylnaphthalene	8.89 × 10^−8^	8.89 × 10^−8^	8.89 × 10^−8^	1.32 × 10^−8^	1.32 × 10^−8^	1.32 × 10^−8^
benzo[a]anthracene	1.68 × 10^−8^	5.63 × 10^−8^	1.42 × 10^−7^	1.49 × 10^−8^	4.97 × 10^−8^	1.25 × 10^−7^
chrysene	2.37 × 10^−10^	9.91 × 10^−10^	2.27 × 10^−9^	2.26 × 10^−10^	9.45 × 10^−10^	2.17 × 10^−9^
benzo[b]fluoranthene	2.61 × 10^−8^	7.17 × 10^−8^	1.70 × 10^−7^	1.74 × 10^−8^	4.78 × 10^−8^	1.13 × 10^−7^
benzo[k]fluoranthene	1.41 × 10^−9^	5.53 × 10^−9^	1.24 × 10^−8^	1.56 × 10^−9^	6.11 × 10^−9^	1.38 × 10^−8^
di-(2-ethylhexyl) phthalate	2.71 × 10^−7^	3.45 × 10^−7^	4.48 × 10^−7^	4.89 × 10^−7^	6.23 × 10^−7^	8.10 × 10^−7^
*∑R*	6.77 × 10^−7^	1.05 × 10^−6^	1.79 × 10^−6^	5.39 × 10^−7^	7.44 × 10^−7^	1.08 × 10^−6^

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
