# Peer review of "Characteristics and Health Risk Assessment of Semi-Volatile Organic Contaminants in Rural Pond Water of Hebei Province"

_ijerph, 2019, doi:10.3390/ijerph16224481_

Round 1

Reviewer 1 Report

This is well done manuscript however there is one major mistake regarding drawn conclusions that need to be corrected. Authors are drawing conclusions on potential carcinogenic and non-carcinogenic risk associated with ingestion of dermal exposure on  pond water contamianed with semivolatile organic contaminant based on WRONG assumtion that it is that the person is exposed for this source of pollution for 365 days per year I understand that this is due to use of EPA formula, but Authors should at least make a comment that they are aware of the fact that the real exposure frequency will not be 365 days of e.g.  "ingesting" of pond water. Please revise the conlucsion section by addressing this issue.

There are also other but minor mistakes within the text mostly regarding grammar or spelling erros:

line 36 there is incomplete sentence, laking verb : '.... sediments that are highly toxic, difficult to decompose, long-distance ..."

line 54- gramatically incorect : " having been prescribed" - should be have been

line 69 " But..." - do not start sentence with But

line 90 please change word "concentated" on located, since sampling sites can not be concentrated

line 93 change PH on pH and change on site into in situ

line 108 change groundwater into water since the groundwater was not examined during this study

line 199-201 re write the sentence since word "detected" was used 3 times in one sentece

line 207- please at least try to name probable origin of this source of pollution

Author Response

The changes we made in the revised manuscript have been highlighted in RED for your comments. The detailes are as follow:

Point 1: This is well done manuscript however there is one major mistake regarding drawn conclusions that need to be corrected. Authors are drawing conclusions on potential carcinogenic and non-carcinogenic risk associated with ingestion of dermal exposure on pond water contamianed with semivolatile organic contaminant based on WRONG assumtion that it is that the person is exposed for this source of pollution for 365 days per year I understand that this is due to use of EPA formula, but Authors should at least make a comment that they are aware of the fact that the real exposure frequency will not be 365 days of e.g.  "ingesting" of pond water. Please revise the conclusion section by addressing this issue. 

Response 1:   We thank the reviewer for the considerable comments. The reviewer is current that we do not have considered comprehensively about the real conditions about the exposure frequency. The comment has been made in line 167 and line 352 on page 4 of the revised manuscript to clarify the assumption.

Point 2: There are also other but minor mistakes within the text mostly regarding grammar or spelling errors:

line 36 there is incomplete sentence, laking verb : '.... sediments that are highly toxic, difficult to decompose, long-distance ..."

line 54- gramatically incorect : " having been prescribed" - should be have been

line 69 " But..." - do not start sentence with But

line 90 please change word "concentated" on located, since sampling sites can not be concentrated

line 93 change PH on pH and change on site into in situ

line 108 change groundwater into water since the groundwater was not examined during this study

line 199-201 re write the sentence since word "detected" was used 3 times in one sentece

line 207- please at least try to name probable origin of this source of pollution

Response 2: The mistakes mentioned by the reviewer have been corrected (highlighted in red colour) accordingly in the revised manuscript. Also, the entire manuscript has been carefully checked to avoid grammar and spelling errors.

I also upload a Word. Please see the attachment

Reviewer 2 Report

Comments in line 34 about references for human risk pollutants.

Line 153 to 165. Equations has some missing information.

Author Response

The changes we made in the revised manuscript have been highlighted in RED for your comments. The detailes are as follow:

Point 1: Line: 34: Author´s could reference information about risk levels of pollutants for human health.

Response 1:   The reference information about risk levels of pollutants for human health is now added in line 73 on page 2 of the revised manuscript.

Point 2: Line: 153: Is It mean that E = Di?

Response 2: The error has been corrected. Please refer to line 172 of the revised manuscript.

Point 3: Line 154: I can´t understand if it a theorical value?.

Response 3:   RfD refers to the oral reference dose from the Integrated Risk Information System (IRIS). We have modified the definition in line 173 of the revised manuscript. There is also citation of the IRIS in line 179 of the original manuscript.

Point 4: Line 160: Authors define HQ(n) for any hazard or the set of specific hazard.

Response 4:   HQn represents the hazard quotient for the nth toxicant. The definition has been clarified now in line 184.

Point 5: Line 164: Authors define HQ(n) for any hazard or the set of specific hazard.

Response 5: Please refer to Response 4.

Point 6:  Comments in line 34 about references for human risk pollutants. Author´s could reference information about risk levels of pollutants for human health.

Response 6:   More reference information has been added in the introduction in line 35 of the revised manuscript.

Point 7:  Line 153 to 165. Equations has some missing information.

Response 7:  The missing information has been added in line 172 and 173 of the revised manuscript.

Reviewer 3 Report

This study analyzed and evaluated 64 semi-volatile organic contaminants in ten rural pond water samples collected in Hebei province, and evaluated the HQ and HI values through direct ingestion pathway and dermal absorption pathway to assess the potential hazards on humans. It is an interesting content, but arranged structure needs to be further improved. Therefore, it needs minor revision before it is published in this journal. The following issues should be carefully addressed.

There are some grammatical errors and the sentences are monotonous in the manuscript, the authors need to go through the entire manuscript sentence by sentence; Authors should clearly mention the novelty of the study in Introduction; Organic pollutants present in pond water was focused in this studied. Actually, heavy metal ions present in pond water also pose health risks to humans, and they may be derived from the dissolution of minerals. In order to complete the environmental impact factors, the authors should introduce the effect of organic pollutants and heavy metal ions present in pond water in the “Introduction”. The following relevant references may be added to the “Introduction” section of your manuscript (Minerals Engineering 137 (2019) 1–9; Minerals Engineering 141 (2019) 105846). Give more detail analysis and discussion in the section of “Discussion”, and it should be amended by including a brief analysis respect to the use of the reported results to other scale; Conclusions should be rearranged.

Author Response

The changes we made in the revised manuscript have been highlighted in RED for your comments. The detailes are as follow:

Point 1: There are some grammatical errors and the sentences are monotonous in the manuscript, the authors need to go through the entire manuscript sentence by sentence.

Response 1:   The manuscript has been carefully revised regarding to the grammatical errors according to the reviewer’s comment.

Point 2: Authors should clearly mention the novelty of the study in Introduction.

Response 2:  The novelty of this study is to fill the blank of the research on SVOCs distribution in the rural pond water and their health risk assessment to the human body. We have highlighted the novelty in the introduction in line 73-85 of the revised manuscript.

Point 3: Organic pollutants present in pond water was focused in this studied. Actually, heavy metal ions present in pond water also pose health risks to humans, and they may be derived from the dissolution of minerals. In order to complete the environmental impact factors, the authors should introduce the effect of organic pollutants and heavy metal ions present in pond water in the “Introduction”. The following relevant references may be added to the “Introduction” section of your manuscript (Minerals Engineering 137 (2019) 1–9; Minerals Engineering 141 (2019) 105846).

Response 3:   The impact of heavy metal has been reviewed in the revised introduction in line 35 and the reference suggested by the reviewer has been cited accordingly as well.

Point 4:  Give more detail analysis and discussion in the section of “Discussion”, and it should be amended by including a brief analysis respect to the use of the reported results to other scale.

Response 4:   Thank you for your valuable advice. The discussion about the use of the reported results to other scale has been carefully revised in line 202-207, 230-235, 243-244, 261-262, 265-271, 293-297 and 331-332.

Point 5: Conclusions should be rearranged.

Response 5: The conclusion has been carefully revised and rearranged according to the reviewer.
